# Early Surgery with Neuraxial Anaesthesia in Patients on Chronic Antiplatelet Therapy with a Proximal Femur Fracture: Multicentric Randomised Clinical Trial

**DOI:** 10.3390/jcm10225371

**Published:** 2021-11-18

**Authors:** Rafael Anaya, Mireia Rodriguez, Angélica Millan, Francesca Reguant, Jordi Llorca, Patricia Guilabert, Ana Ruiz, Percy-Efrain Pantoja, José María Gil, Victoria Moral, Angela Merchán-Galvis, Maria Jose Martinez-Zapata

**Affiliations:** 1Anesthesiology Service, Hospital de la Santa Creu i Sant Pau, 08025 Barcelona, Spain; rafaelanaya30@yahoo.es (R.A.); mrodriguezpr@santpau.cat (M.R.); jgils@santpau.cat (J.M.G.); vmoralg@santpau.cat (V.M.); 2Orthopedic and Traumatology Surgery Service, Hospital de la Santa Creu i Sant Pau, 08025 Barcelona, Spain; amillan@santpau.cat; 3Anesthesiology Service, Xarxa Assitencial Universitària de Manresa, 08243 Barcelona, Spain; freguant@althaia.cat (F.R.); jllorca@althaia.cat (J.L.); 4Anesthesiology Service, Hospital de la Vall d’Hebron, 08035 Barcelona, Spain; patricia.guilabert@gmail.com; 5Anesthesiology Service, Hospital Clinic de Barcelona, 08036 Barcelona, Spain; anaruiz@clinic.cat; 6Public Health and Clinical Epidemiology Service-Iberoamerican Cochrane Centre, IIB Sant Pau, 08025 Barcelona, Spain; efrapb182@gmail.com (P.-E.P.); amerchang@santpau.cat (A.M.-G.); 7Departamento de Medicina Social y Salud Familiar, Universidad del Cauca, Popayan 190003, Colombia; 8Centro de Investigación Biomédica en Red de Epidemiología y Salud Pública (CIBER of Epidemiology and Public Health), 28029 Madrid, Spain

**Keywords:** femur fracture, antiplatelet drugs, randomized clinical trial

## Abstract

Background: Patients with proximal femur fracture on antiplatelet treatment benefit from early surgery. Our goal was to perform early surgery under neuraxial anaesthesia when indicated by the platelet function test. Methods: We conducted a multicentre randomised open-label parallel clinical trial. Patients were randomised to either early platelet function-guided surgery (experimental group) or delayed surgery (control group). Early surgery was programmed when the functional platelet count (as measured by Plateletworks) was >80 × 10^9^/L. The primary outcome was the emergency admission-to-surgery interval. Secondary outcomes were platelet function, postoperative bleeding, medical and surgical complications, and mortality. Results: A total of 156 patients were randomised, with 78 in each group, with a mean (SD) age of 85.96 (7.9) years, and 67.8% being female. The median (IQR) time to surgery was 2.3 (1.5–3.7) days for the experimental group and 4.9 (4.4–5.6) days for the control group. One-third of patients did not achieve the threshold functional platelet count on the first day of admission, requiring more than one test. There was no difference in clinical outcomes between groups. Conclusions: A strategy individualised according to the platelet function test shortens the time to proximal femur fracture surgery under neuraxial anaesthesia in patients on chronic antiplatelet treatment. Better powered randomised clinical trials are needed to further evaluate the clinical impact and safety of this strategy.

## 1. Introduction

In industrialised countries, most proximal femur fractures occur in elderly patients, accounting for approximately 42–50% of all fractures [1]. Following fracture, mortality at 30 days is around 10% (9–11%), while mortality at 360 days ranges widely between 18% and 33%, with elderly people, frail people, males, and patients with dementia at higher risk [2,3,4,5]. Increased life expectancy and an ageing population further aggravate hip fracture incidence as a major public health problem in terms of morbidity, mortality, functional impact, and cost from both a provider and societal perspective [6,7].

Associated comorbidities and chronic treatments increase the complexity of perioperative management of elderly patients. It is not uncommon for hip fracture patients to be receiving chronic antiplatelet treatment, which lengthens hospital stay.

Early surgery, i.e., within less than 48 h, has been reported to decrease morbidity and mortality, hospital stay, and perioperative complications associated with femur fractures [8,9]. Even patients on antiplatelet treatment seem to experience reduced mortality and hospital stay from early surgery under general anaesthesia [10,11,12].

Some studies have reported that neuraxial anaesthesia compared with general anaesthesia reduces hospital stay and mortality associated with hip fracture surgery [13,14]. Nonetheless, there is some controversy regarding the comparative benefits of general and neuraxial anaesthesia; clinical trials are ongoing to assess which of the two types of anaesthesia is more effective and safer [15,16], while another ongoing clinical trial is assessing which of these anaesthetic techniques may have a greater impact on reducing delirium in elderly patients with hip fractures [17].

In clinical practice, there is no consensus regarding the best type of anaesthesia for patients with femur fractures on antiplatelet treatment due to the potential risk of bleeding and associated complications, such as epidural hematoma related to neuraxial anaesthesia [18,19].

Before surgery under neuraxial anaesthesia, it is recommended to stop antiplatelet therapy for 3–7 days, depending on the drug type [20,21,22,23]. However, platelet function measurement could reduce this interval and customise it to each patient [24,25]. Our hypotheses were that a strategy for measuring platelet function could reduce the time to surgery under neuraxial anaesthesia for patients on chronic antiplatelet treatment and, consequently, reduce postoperative complications, hospital stay, and costs. We therefore conducted a randomised clinical trial (RCT) to evaluate these hypotheses in adult patients on antiplatelet treatment with a diagnosis of proximal femur fracture.

## 2. Experimental Section

### 2.1. Materials and Methods

This paper was prepared following the Consolidated Standards of Reporting Trials (CONSORT) guidelines, and the clinical trial was approved by the Spanish Agency of Medicines and Medical Devices (AEMPS) and by the ethics committees of the participating hospitals, with the Hospital de la Santa Creu i Sant Pau’s (Barcelona, Spain) as the ethics committee of reference. Written informed consent was obtained from all subjects participating in the trial. The trial was registered on 27 July 2017 prior to patient enrolment at clinicaltrials.gov (NCT03231787; accessed data 17 November 2021).

### 2.2. Study Design

For this multicentre randomised open-label parallel clinical trial, we recruited patients with a proximal femur fracture on chronic antiplatelet therapy who underwent surgery between 26 September 2017 and 5 December 2020. The protocol of this study has been published elsewhere [26].

### 2.3. Participants

Location and eligibility. Four Spanish hospitals participated in this study. According to the trial protocol, inclusion criteria were male and female adults ≥18 years of age; admitted to emergency rooms and diagnosed with a proximal femur fracture; and receiving treatment with antiplatelet agents, such as cyclooxygenase inhibitors (acetylsalicylic acid (ASA) > 100 mg/day or triflusal > 300 mg/day) or P2Y12 receptor inhibitors (any dose of clopidogrel, prasugrel, ticagrelor, or ticlopidine), who underwent surgery with neuraxial anaesthesia after signing a written informed consent. The usual clinical practice of each hospital was followed; thus, as hospitals number #3 and #4 do not delay surgery for patients on ASA, they excluded those patients. Furthermore, also excluded were patients with multiple or pathological fractures, under current management with vitamin K antagonists or new oral anticoagulants, and/or with congenital or acquired coagulopathies.

Interventions. Included patients were randomised to either early platelet function-guided surgery (hereinafter, early surgery) with neuraxial anaesthesia (experimental group) or delayed surgery with neuraxial anaesthesia (control group).
Experimental group. Platelet function was measured on emergency room admission. The threshold indicating candidacy for early surgery was set to a minimum of 80 × 10^9^/L to ensure anaesthetic and surgical safety [27]. In patients with >80 × 10^9^/L functional platelets, surgery was scheduled for within 24 h. In patients with ≤80 × 10^9^/L functional platelets, platelet function was measured daily to check if the minimum threshold count was reached; if platelet count had not normalised by the third day, surgery with neuraxial anaesthesia was scheduled following the margin of safety established for each specific antiplatelet drug (as specified for the control group).Control group. A platelet function test was performed 24 h prior to surgery, blinded so as not to influence clinical decision making regarding the control patients. Surgery with neuraxial anaesthesia was performed following the established margin of safety for each specific antiplatelet drug: 3 days for ASA > 100 mg/day and triflusal > 300 mg/day, 5 days for clopidogrel and ticagrelor, 7 days for prasugrel, and 10 days for ticlopidine [20,21,22,23].

Platelet function. Within 10 min of blood extraction, platelet function was measured, using Plateletworks (Helena Laboratories, Beaumont, TX, USA), in a 2-step system that determines the functional platelet count: (1) the total number of platelets was counted in the first blood sample containing ethylenediaminetetraacetic acid (EDTA), and (2) the total number of non-functional platelets was counted in the second blood sample containing a platelet agonist (adenosine diphosphate (ADP) for P2Y12 receptor inhibitors or arachidonic acid (AA) for cyclooxygenase inhibitors). The difference between the 2 counts yielded the number of functional platelets [28]. For patients on dual antiplatelet therapy, the agonist used was ADP.

Surgery. The surgical implant decision in orthopaedic treatment depended on fracture type and patient functional status. For non-displaced or impacted subcapital femur fractures (Garden I or II), osteosynthesis was performed using 6.5 mm cannulated screws. For displaced subcapital femur fractures, the surgery type depended on the patient: for non-wandering patients older than 90 years, arthroplasty using an uncemented monopolar hemiarthroplasty; for wandering patients/patients younger than 80 years, hybrid total hip arthroplasty with cemented stem; and for wandering patients older than 80 years, bipolar hemiarthroplasty with cemented stem. Patients with extra-articular fractures underwent osteosynthesis using dynamic hip screws for basicervical or stable pertrochanteric fractures, short endomedullary nails (170 or 180 mm) for unstable pertrochanteric fractures, and long endomedullary nails for subtrochanteric fractures.

Each orthopaedic surgeon followed their hospital’s surgical protocol for the fracture type, and each hospital applied its own transfusion protocol. All patients received antithrombotic and antibiotic prophylaxis drugs in the perioperative period.

### 2.4. Outcomes

The primary outcome was the admission-to-surgery interval, recorded in hours but reported in days for better interpretation. Secondary outcomes were clinical and related to the perioperative period: number of days from emergency admission to 30 days (±15 days) after hospital discharge, platelet function, postoperative bleeding, red blood transfusions (from the data of the Blood Bank of Catalonia, Spain, one unit had a medium volume of 289 (±25) mL), medical and surgical complications, and mortality. We calculated postoperative bleeding based on haemoglobin balance according to equations described by Nadler et al. [29].

### 2.5. Sample Size

An administrative historical database belonging to a participating hospital indicated that the mean (SD) admission-to-surgery interval was 2.8 (3.2) days for in-patients not on antiplatelet treatment and without coagulopathies. The sample size needed to detect a decrease of ≥1.5 days in the admission-to-surgery interval was therefore estimated at 78 patients per group, for a 2-sided contrast alpha risk of 5%, beta risk of 20%, and 10% loss rate (GRANMO calculator v.7.10 (June 2010), available at: https://www.imim.es/ofertadeserveis/en_granmo.html; accessed data 17 November 2021).

### 2.6. Random Sequence Generation and Allocation Concealment

A computer-generated permuted block randomisation sequence, stratified by hospital, was used for randomisation purposes. A web-based platform (http://www.clinapsis.com; accessed data 17 November 2021) allocated patients 1:1 to the study arms, randomly assigning a numerical code to each patient and the corresponding intervention. Researchers were blinded to the allocation sequence of the study interventions.

### 2.7. Implementation

No predetermined limit was established for the total number of patients recruited per hospital. Recruitment stopped once the number of patients established for the study was achieved. Patients, selected after emergency room admission, were screened for inclusion and exclusion criteria, informed of the aims and purpose of the study, enrolled after signing the informed consent, and randomised to the experimental or control group. Blood was sampled (as described above) to determine platelet function.

Adverse events. A notification system was established for adverse events (AEs), including serious AEs, defined as any AE that caused death, was life-threatening, required hospitalisation, prolonged existing hospitalisation, or caused permanent or significant disability, or an abnormality or congenital malformation.

Data collection and monitoring. All data collected were uploaded to a password-protected electronic web-based database (http://www.clinapsis.com; accessed data 17 November 2021). The study was monitored by the Spanish Clinical Research Network (SCReN). All primary study data were obtained from a review of electronic medical records and direct patient interviews (in person or by telephone).

### 2.8. Blinding

Once patients were randomised, they and the researchers (emergency room clinicians and surgeons) knew to which intervention group patients had been allocated. Researchers were blinded to platelet function status for the control group to avoid influencing decisions regarding time to surgery.

### 2.9. Statistical Analysis

Primary outcome. Data for the primary outcome—the admission-to-surgery interval in days for each study arm—were reported as median and interquartile range (IQR), and the test used was the non-parametric Wilcoxon–Mann–Whitney (Mann–Whitney U) test. Patients who did not undergo surgery were excluded from the main analysis, which was per protocol (randomised patients who underwent surgery). Mortality was also analysed by intention to treat (considering all randomised patients).

Baseline characteristics. Used for the categorical variables, and depending on the contingency table characteristics, was the chi-square test, and the Fisher exact test when there were fewer than 10 observations per cell. Mean (SD) or median (IQR) values were reported for the continuous variables. When distributions were normal, the two-tailed *t*-test was used, and the Mann–Whitney U test as the non-parametric option.

## 3. Results

We screened 2626 patients undergoing surgery due to proximal femur fracture, of whom, 2470 were excluded: 2286 because they did not meet the inclusion criteria (1049 patients because they were taking cyclooxygenase inhibitors—mainly ASA 100 mg/day—at doses below the inclusion criteria and/or anticoagulants). For logistical reasons, 184 patients receiving antiplatelet drugs were not included, as the time to the last dose was above 24 h and 48 h for those on a cyclooxygenase inhibitor or a P2Y12 receptor inhibitor, respectively.

A total of 156 patients (mean (SD) age 85.96 (7.9) years; 67.8% female) were included and randomised to the experimental group (early surgery) or the control group (delayed surgery), 78 per arm. Of the randomised patients, 75 and 68 underwent early surgery and delayed surgery, respectively. Figure 1 depicts the recruitment flowchart, detailing the reasons for losses.

The most frequent medical histories were arterial hypertension (82.5%), surgery (66.4%; principally orthopaedic, digestive, gynaecological, or urological interventions), ischaemic neurological disease (47.5%), diabetes (34.3%), and delirium and cognitive impairment (30.8%). The American Society of Anesthesiologists score was 3 in around 80% of patients who underwent surgery. There were no differences in the frequency of intracapsular and extracapsular fractures. The indication of all included drugs was to prevent cardiovascular ischemic events. ASA was prescribed at a dose between 150 and 300 mg/day, and the most frequent indication was vascular cerebral ischemia (15 patients); other indications were auricular fibrillation (5 patients), coronary ischemia (1 patient) and arterial peripheral ischemia (2 patients). Clopidogrel was the most used antiplatelet drug (79.0% of patients) (Table 1).

### 3.1. Platelet Function

There were no statistically significant between-group differences in functional platelet counts at the time of the last measurement before surgery (Table 1).

In the experimental group, 67 patients (89.3%) were scheduled for early surgery based on a >80 × 10^9^/L functional platelet threshold. Most patients (65.3%) had >80 × 10^9^/L functional platelets at the first test, which was performed a median (IQR) of 0.8 (0.6–1.6) days after admission and a median (IQR) of 1.9 (0.9–2.9) days after the last antiplatelet dose, with no differences between cyclooxygenase and P2Y12 receptor inhibitors (Table 2). However, 24% of patients needed two or three tests before achieving the threshold functional platelet count, with eight patients (10.7%) assigned to the delayed surgery group (see Table 2). Stratified by antiplatelet drug groups, the threshold functional platelet count was achieved at the first test by 81.2% and 73.1% of patients on cyclooxygenase inhibitors and on P2Y12 receptor inhibitors, respectively.

### 3.2. Primary Outcome

The median (IQR) admission-to-surgery interval was 2.3 (1.5–3.7) days for the experimental group versus 4.9 (4.4–5.6) days for the control group (*p* < 0.001), results that were maintained when the analysis was stratified by antiplatelet drug groups: for cyclooxygenase inhibitors, 2.0 (1.2–3.7) days for the experimental group versus 4.5 (3.74–4.7) days for the control group (*p* = 0.02) and, for P2Y12 receptor inhibitors, 2.4 (1.6–3.5) days for the experimental group versus 4.9 (4.5–5.7) days for the control group (*p* < 0.001) (Figure 2).

The linear regression coefficients did not point to any strong effect of clinical factors on the primary outcome, although time to surgery was influenced by the different participating hospitals. The early surgery strategy consistently reduced time to surgery by 2.4 days independently of other factors (Appendix A).

### 3.3. Secondary Outcomes

The most frequent orthopaedic treatment was osteosynthesis (57.3%), and most patients (89.5%) underwent neuraxial anaesthesia. Haemoglobin values decreased after surgery, but there were no differences between the groups except for the measurement 2 h after surgery. Perioperative blood loss was similar between groups (Table 3). Hospital stays were shorter for the experimental group compared to the control group, at median (IQR) 9.7 days (8.2–13.4) versus 12.6 (10.1–16.5) days (*p* < 0.001).

There were no differences in perioperative complications or mortality between the groups (Table 4). Nearly three-quarters (70.6%) of patients received at least one unit of red blood cell transfusion. No spinal haematomas occurred. There were no differences in the number of AEs or serious AEs between the groups during follow-up; 61.7% of AEs were resolved, and 38.39% ended in death (Appendix A).

## 4. Discussion

The results of this RCT point to platelet function monitoring as an effective strategy to reduce the admission-to-surgery interval when neuraxial anaesthesia is used for elderly patients on chronic antiplatelet treatment admitted with femur fractures. The early surgery strategy compared to delayed surgery reduced the admission-to-surgery interval by 2 days (median 2.4 days for the early surgery group), without increasing perioperative AEs and complications, and reduced hospital stays by 3 days.

Note that a number of meta-analyses comparing early and delayed surgery have included studies that exceed the limit of 48 h for early surgery (intervals up to 5 days), yet report benefits [10,11,12]. Some guidelines recommend surgery in the first 24–48 h after admission, as early surgery reduces morbidity, mortality, hospital stay, and perioperative complications [8,9]. General anaesthesia must be used for early surgery in patients in whom chronic antiplatelet treatment has not been discontinued within the established safety period, given the high risk of spinal haematoma from neuraxial anaesthesia [8,9,10,11,12].

To minimise the risk of spinal haematoma and surgical bleeding [27], we chose the functional platelet count of >80 × 10^9^/L as a threshold that would optimise safety in adults on antiplatelet drugs who were candidates for surgery under neuraxial anaesthesia, especially complex in elderly patients, and sometimes requiring various attempts. This threshold was achieved, on average, 0.8 days from admission and 1.9 days from the last administered antiplatelet dose.

Around one-half of our patients could theoretically have undergone surgery under neuraxial anaesthesia in the first 24 h based on achieving the threshold functional platelet count. However, due to a lack of surgery capacity in some of the study hospitals, the real admission-to-surgery interval was double that period (2 days). The interval would therefore undoubtedly have been shorter if better logistical planning had been implemented.

Around 25% of our patients needed a second or third test to achieve the threshold functional platelet count, while around 10% of patients from the early surgery group were assigned to delayed surgery, either because they failed to reach the threshold after three tests or because their blood sample was non-analysable. These results suggest that some patients in treatment with antiplatelet agents take more time to recover haemostasis and, if not previously identified, may experience more surgical bleeding and perioperative complications. Therefore, we propose individualising surgery strategies according to platelet function test results, as it means that neuraxial anaesthesia, associated with fewer complications, can be used [13,14,15,16,17].

While neuraxial anaesthesia was indicated for all our patients, some, however, underwent general anaesthesia because of a change in their condition immediately before surgery. There were no complications related to neuraxial anaesthesia in the first 30 days of follow-up after surgery. We found no differences between groups in haemoglobin concentrations in functional platelets before surgery. Although the early surgery group showed significantly lower haemoglobin concentrations 2 h after surgery, there was no correlation with the number of postoperatively transfused red blood cells. There were significant differences in neither postoperative complications nor perioperative mortality. Our patients will be followed up until 12 months from surgery to observe if these results are maintained over the longer term.

The main strength of our study is that it addresses the important problem of anaesthesia for patients on antiplatelet treatment with femur fractures. In addition, it was well powered for the primary outcome, and the fact that randomisation was stratified by hospital ensured a pragmatic trial design, as hospitals were included with different clinical practices regarding delayed surgery for patients on chronic antiplatelet treatment. While in two hospitals (#3 and #4), ASA was not a criterion to delay surgery, this criterion did not influence the results, because randomisation and allocation were stratified by hospital. In the analysis by antiplatelet group (cyclooxygenase inhibitors versus P2Y12 receptor inhibitors), we therefore continued to find consistently significant differences in the admission-to-surgery intervals between the early and delayed surgery groups.

As for limitations, our study sample size was not sufficient to determine differences for adverse outcomes. We screened a large number of patients because our inclusion criteria were very open (adults > 18 years old); however, only 13% of screened patients were included because most potential candidates were on ASA at low doses (100 mg). Another important limitation, as well as a source of between-hospital heterogeneity in the execution of the study, was the lack of availability of operating rooms for surgery once a patient achieved the threshold functional platelet count. This would indicate, therefore, that there is still substantial scope to optimise the admission-to-surgery interval.

## 5. Conclusions

Patients on chronic antiplatelet treatment admitted for proximal femur fracture and requiring surgery with neuraxial anaesthesia can benefit from early surgery if the admission-to-surgery interval is shortened by surgery scheduling based on platelet function testing. Individualising the surgery timing decision is especially indicated for patients with lower haemostasis recovery rates and at a higher risk of surgical bleeding and associated complications. Our findings need, however, to be confirmed by further RCTs of a size sufficient to detect potential differences in clinical outcomes.

## Figures and Tables

**Figure 1 jcm-10-05371-f001:**
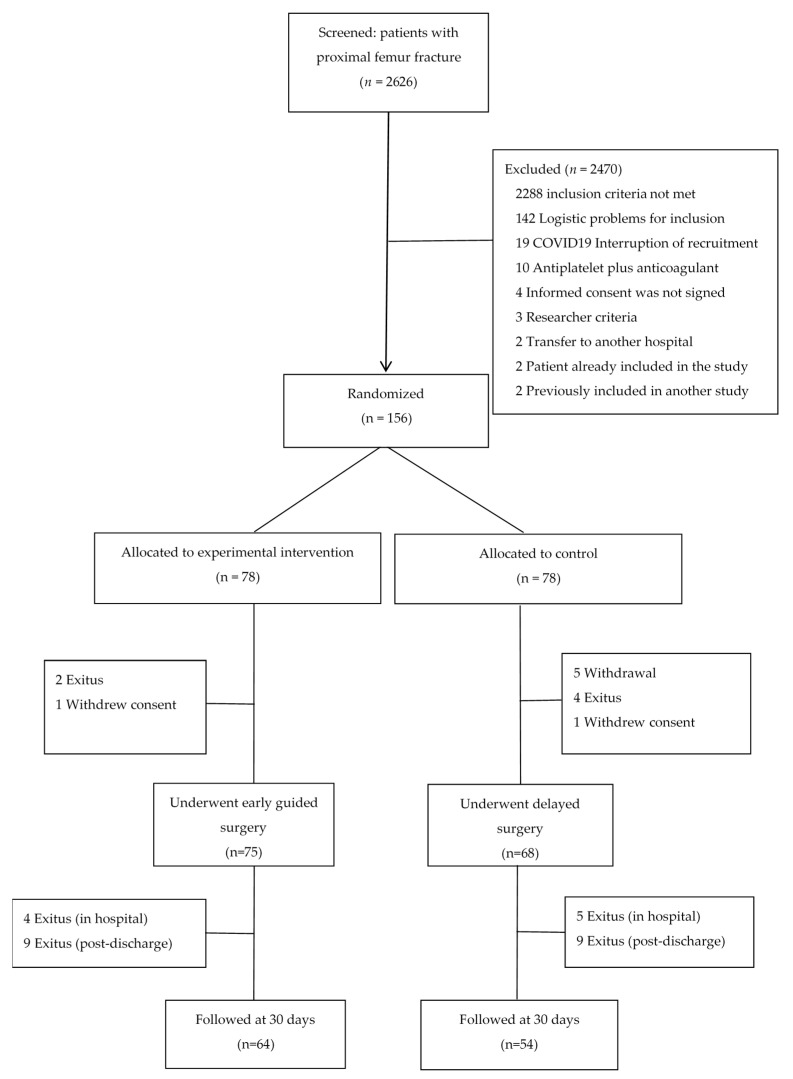
Flow chart of patients.

**Figure 2 jcm-10-05371-f002:**
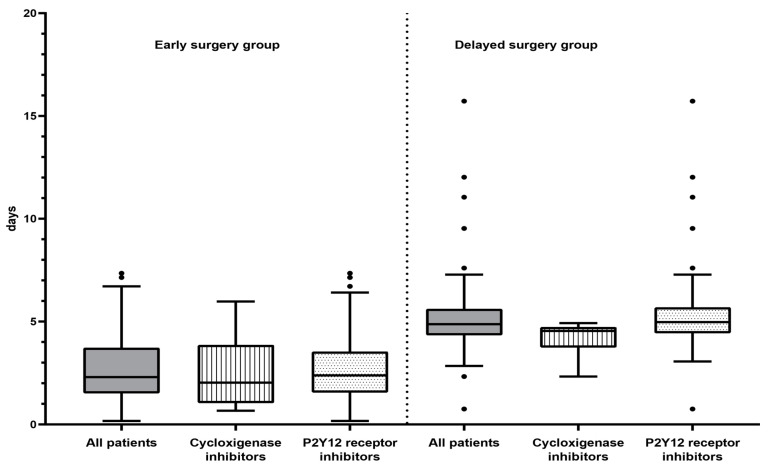
Admission-to-surgery interval (days) by study group and antiplatelet drug group. • Outlier value.

**Table 1 jcm-10-05371-t001:** Per-protocol baseline characteristics of patients by study arm.

	Early Surgery *N* = 75 *n* (%)	Delayed Surgery *N* = 68 *n* (%)	*p*
Age [m(SD)]	85.44 (8.7)	86.53 (6.8)	0.4
Female	55 (73.3)	42 (61.8)	0.13
Medical history			
Arterial hypertension	61 (81.3)	57 (83.8)	0.69
Surgery	55 (68.0)	40 (52.9)	0.06
Ischaemic neurological disease	34 (45.3)	34 (50.0)	0.57
Delirium and cognitive impairment	19 (25.3)	25 (36.7)	0.14
Diabetes	28 (37.3)	21 (30.9)	0.42
Coronary disease	31 (41.3)	20 (29.4)	0.14
Chronic renal insufficiency	24 (32.0)	17 (25.0)	0.35
Pulmonary disease	15 (20.0)	13 (19.1)	0.89
Oncological disease	9 (12.0)	9 (13.2)	0.824
Auricular fibrillation	11 (14.6)	7 (10.3)	0.43
Other	27 (36.5)	25 (36.8)	0.97
American Society of Anesthesiologists score			
1	1 (1.3)	0 (0)	0.37
2	3 (4.0)	6 (8.8)	
3	67 (89.3)	57 (83.8)	
4	4 (5.3)	5 (7.3)	
Type of femur fracture			
Intracapsular	34 (45.3)	37 (54.4)	0.28
Extracapsular	41 (54.7)	31 (45.6)	
Antiplatelet drug:			
Clopidogrel	59 (78.7)	54 (79.4)	0.87
ASA > 100 mg	13 (17.3)	10 (14.7)	
Clopidogrel/ASA	2 (2.7)	3 (4.4)	
Cilostazol/triflusal > 300 mg	0 (0)	1 (1.5)	
Ticagrelor	1 (1.3)	0 (0)	
Antiplatelet drug based on action mechanism			
Cyclooxygenase inhibitors	13 (17.3)	11 (16.2)	0.85
(AAS > 100 and triflusal > 300)			
P2Y12 receptor inhibitors	62 (82.7)	57 (83.8)	
(clopidogrel/ticagrelor)			
Functional platelet count * (×10^9^/L)	***N* = 67**	***N* = 66**	
All patients	118 (90–144)	136.5 (98–176)	0.13
On cyclooxygenase inhibitors	117 (88–144)	108 (68–166)	0.92
(11/11) †			
On P2Y12 receptor inhibitors	118 (91.5–146)	140 (99–192)	0.08
(56/55) †			

* Pre-surgery values expressed as median (IQR). Missing data are due to patients not reaching the functional platelet threshold after 3 tests or having non-analysable samples. † Numbers in the early and delayed surgery groups, respectively.

**Table 2 jcm-10-05371-t002:** Pre-surgery functional platelet tests for the experimental group.

	Early Surgery *N* = 75	Last Antiplatelet Dose to Platelet Count > 80 × 10^9^/L	Admission to Platelet Count > 80 × 10^9^/L
	*n* (%)	Days, Median (IQR)	Days, Median (IQR)
All patients	67 (89.3)	1.9 (0.9–2.9)	0.8 (0.6–1.6)
On cyclooxygenase inhibitors	11 (14.66)	1.97 (1.89–2.04)	0.70 (0.65–1.11)
On P2Y12 receptor inhibitors	56 (74.66)	1.92 (0.96–2.17)	0.88 (0.61–1.60)
Platelet function tests (*n*)			
1	49 (65.3)	1.9 (0.9–2.0)	0.7 (0.4–0.9)
2	13 (17.3)	1.9 (1.9–2.9)	1.6 (1.40–1.8)
3	5 (6.7)	2.9 (2.9–3.0)	2.7 (2.7–3.3)
Referred for delayed surgery *	8 (10.7)	-	-

* Delayed surgery decision was due to not reaching the functional platelet threshold after 3 tests (*n* = 4) or having a non-analysable blood sample (*n* = 4). These patients were analysed at the early surgery group.

**Table 3 jcm-10-05371-t003:** Perioperative secondary outcomes.

	Early Surgery *N* = 75 *n* (%)	Delayed Surgery *N* = 68 *n* (%)	*p*
Orthopaedic treatment			0.31
Arthroplasty	29 (38.7)	32 (47.1)	
Osteosynthesis	46 (62.2)	36 (52.9)	
Anaesthesia type			0.17
General +/− peripheral nerve block	5 (6.7)	10 (14.7)	
Neuraxial +/− peripheral nerve block	70 (93.3)	58 (85.3)	
Tranexamic acid used	3 (4.0)	0 (0)	0.09
Surgery duration (median (IQR) mins; *t*-test)	70 (46–87)	64 (52–95.5)	0.77
Haemoglobin values (mean (SD) g/L)			
On admission (*n* = 143)	117.2 (18.7)	119.5 (16.0)	0.42
12 h pre-surgery (*n* = 57/60)	109.9 (14.3)	108.7 (15.7)	0.67
2 h post-surgery (*n* = 45/40)	104.8 (14.2)	114.0 (14.7)	0.004
24 h post-surgery (*n* = 71/63)	96.1 (11.7)	97.2 (15.1)	0.64
5 d post-surgery (*n* = 67/61)	98.3 (12.8)	98.7 (12.6)	0.85
Perioperative blood loss	577.3	604.2	0.94
(median (IQR) mL) (*n* = 73/67)	(1.2–1061.5)	(1.7–1020.0)	
Hospital stay	9.7	12.6	<0.001
(median (IQR) days) (*n* = 71/63)	(8.2–13.4)	(10.1–16.5)	

**Table 4 jcm-10-05371-t004:** Perioperative complications and mortality.

	Early Surgery *N* = 75 *n* (%)	Delayed Surgery *N* = 68 *n* (%)	*p*
Transfusions			
Patients with at least 1 transfusion	53 (70.7)	48 (70.6)	0.99 0.45
Patients by number of transfusions			
1	35 (46.7)	39 (57.3)	
2	11 (14.7)	7 (10.3)	
3	5 (6.7)	2 (2.9)	
4	2 (2.7)	0 (0)	
Units transfused post-surgery			
Mean (SD)	1.68 (1.6)	1.46 (1.3)	0.37 0.13
Units transfused			
1	17 (22.7)	14 (20.6)	
2	19 (25.3)	27 (39.7)	
3–6	17 (22.7)	7 (10.3)	
Complications in hospital	***N* = 75**	***N* = 68**	
Wound: all patients	6 (8.0)	8 (11.8)	0.58
Infection	0 (0)	1 (1.5)	-
Hematoma	6 (8.0)	8 (11.8)	0.58
Re-intervention	0 (0)	1 (1.5)	-
Medical: all patients	41 (54.7)	39 (57.3)	0.87
Urinary tract infection	16 (21.3)	12 (17.6)	0.58
Delirium	10 (13.3)	11 (16.2)	0.63
Acute renal insufficiency	4 (5.3)	9 (13.2)	0.14
Symptomatic hypotension	3 (4.0)	7 (10.3)	0.19
Pressure ulcer	2 (2.7)	6 (8.8)	0.15
Cardiac insufficiency	3 (4.0)	4 (5.9)	0.71
Death	4 (5.3)	5 (7.3)	0.74
Pneumonia	1 (1.3)	3 (4.4)	0.35
Ictus	2 (2.7)	1 (1.5)	1
Spinal haematoma	0	0	-
Other	19 (25.3)	18 (26.5)	0.88
Complications 30 ± 15 d post-discharge	***N* = 71**	***N* = 63**	
Wound: all patients	2 (2.8)	2 (3.2)	-
Haematoma	1	0	-
Dislocation	1	2	-
Medical: all patients	19 (25.3)	21 (34.9)	0.23
Mortality	9 (12.7)	9(14.3)	0.8
Urinary tract infection	4 (5.6)	3 (4.8)	0.82
Pneumonia	1 (1.4)	3 (4.8)	0.34
Delirium	0 (0)	2 (3.2)	0.22
Myocardial infarction	2 (2.8)	0 (0)	0.5
Gastrointestinal bleeding	3 (4.2)	1 (1.6)	0.62
COVID-19	3 (4.2)	0 (0)	0.25
Other	9 (12.7)	16 (25.4)	0.08
Perioperative mortality (from randomisation to 30 ± 15 d post-discharge)			
ITT analysis *	15 (19.2)	18 (23.1)	0.56
PP analysis *	13 (17.3)	14 (20.6)	0.62

* ITT, intention to treat, all randomised patients were analysed; PP, per protocol, only randomised patients who underwent surgery were analysed.

## Data Availability

Data of the study are not publicly available, but they can be requested with previous justification to the authors.

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
