# Peer review of "Early Surgery with Neuraxial Anaesthesia in Patients on Chronic Antiplatelet Therapy with a Proximal Femur Fracture: Multicentric Randomised Clinical Trial"

_jcm, 2021, doi:10.3390/jcm10225371_

Round 1

Reviewer 1 Report

M&M:

Why did you include patients >18? The mean age of your study population is >85 years, for usual inclusion in the geriatric population starts at 65 / 70 years.

Results:

Figure 1 is insufficient, please revise.

Table 1: Please specify type of fracture (e.g. intertrochanteric).

Please specify indication for the application of the individual drug (e.g. why ASA >100 mg?)

In M&M, you state that "Platelet function was measured on emergency room admission"; why is the median for the first test 0.8 days after admission?

How did you assign the 8 patients of initial early surgery group, that were sent for delayed surgery group after insufficient platelet count: changed to delayed surgery group?

Table 3: Please specify orthopedic treatment section

Please specify exact measurement of preoperative blood loss in M&M section.

Please specify the volume of 1 blood unit in M&M.

Discussion:

Almost 80% of the patients took clopidogrel; according to the mentioned margin of safety, surgery was performed after 5 days in the control group. Please present and discuss current literature, pointing out the matter of early surgical care (<48 or even <24h from admission) and its influence on selection of anesthesiologic procedures.

Please revise the sentence in line 314, it is unclear.

Author Response

Review Report Form 1

Thanks very much for the comments. We have revised English language.

Comments and Suggestions for Authors

M&M:

Why did you include patients >18? The mean age of your study population is >85 years, for usual inclusion in the geriatric population starts at 65 / 70 years.

Answer: We included a wide inclusion criterion to not exclude any patient with femur fracture and antiplatelet drugs. In view of the results, it would have been more feasible to establish a higher threshold for age.

Results:

Figure 1 is insufficient, please revise.

Answer: We have rectified Figure 1-

Table 1: Please specify the type of fracture (e.g. intertrochanteric).

Answer: We collected the data following the fracture classification of intra and extracapsular. The specific type of fracture intertrocanteric, … was not collected.

Please specify indication for the application of the individual drug (e.g. why ASA >100 mg?)

Answer: We have specified the following in results:

“The indication of all included drugs was for prevent cardiovascular ischemic events. ASA was prescribed at a dose between 150 and 300 mg/day and the most frequent indication was vasculocerebral ischemia (15 patients), other indications were auricular fibrillation (5 patients), coronary ischemia (1 patient) and arterial peripheral ischemia (2 patients).”

In M&M, you state that "Platelet function was measured on emergency room admission"; why is the median for the first test 0.8 days after admission?

Answer: Some of the patients were admitted outside working hours (in the afternoon or at night) and they were recruited into the study the next morning.

How did you assign the 8 patients of the initial early surgery group, that were sent for the delayed surgery group after insufficient platelet count: changed to delayed surgery group?

Answer: They received the delayed surgery. These patients were analyzed in the group that they were assigned at the randomization (ITT principle). We added in the note of Table 2 “They were analysed at the early surgery group.”

Table 3: Please specify orthopedic treatment section

Answer: We have specified the following in methods:

“The surgical implant decision in orthopaedic treatment depended on fracture type and patient functional status. For non-displaced or impacted subcapital femur fractures (Garden I or II), osteosynthesis was performed using 6.5-mm cannulated screws. For displaced subcapital femur fractures, surgery type depended on the patient:  for non-wandering patients older than 90 years, arthroplasty using an uncemented monopolar hemiarthroplasty; for wandering patients/patients younger than 80 years, hybrid total hip arthroplasty with cemented stem; and for wandering patients older than 80 years, bipolar hemiarthroplasty with the cemented stem. Patients with extra-articular fractures underwent osteosynthesis using dynamic hip screws for basicervical or stable pertrochanteric fractures, short endomedullary nails (170 or 180mm) for unstable pertrochanteric fractures, and long endomedullary nails for sub-trochanteric fractures.”

Please specify exact measurement of preoperative blood loss in M&M section.

Answer: we have added in methods “We calculated postoperative bleeding based on haemoglobin balance according to equations described by Nadler et al. [29].”

And we have also included the reference.

Please specify the volume of 1 blood unit in M&M.

Answer: we have added in methods

“…red blood transfusions (from data of Blood Bank of Catalonia, Spain, one unit had a medium volume of 289 (± 25) mL), ..”

Discussion:

Almost 80% of the patients took clopidogrel; according to the mentioned margin of safety, surgery was performed after 5 days in the control group. Please present and discuss current literature, pointing out the matter of early surgical care (<48 or even <24h from admission) and its influence on selection of anesthesiologic procedures.

Answer: we have added in discussion:” General anaesthesia must be used for early surgery in patients in whom chronic antiplatelet treatment has not been discontinued within the established safety period, given the high risk of spinal haematoma from neuraxial anaesthesia [8-12].”

Please revise the sentence in line 314, it is unclear.

Answer: we have modified to:

”We screened a large number of patients because our inclusion criteria were very open (adults >18 years old); however, only 13% of screened patients were included because most potential candidates were on ASA at low doses (100 mg).”

Reviewer 2 Report

Dear Author, 

This manuscript aims to assess whether patients with proximal femur fracture on chronic antiplatelet therapy may benefit from early surgery by shortening of the admission-to-surgery interval. 

Introduction is thorough and it highlights current principles of perioperative antiplatelet therapy management in a certain group of patients.

Experimental section elaborates on every step of the study Including operative techniques and types of anesthesia.

Results comprise adequate graphs, but paging could be improved.

Discussion should include another study limitation, which is management of pediatric population potentially included in this field of interest. Here is an article about Outcome of pinning in patients with slipped capital femoral epiphysis: risk factors associated with avascular necrosis, chondrolysis, and femoral impingement, published in J Int Med Res, DOI 10.1177/0300060517731683.

This article brings valuable information in a certain field of interest. Shortening the hospital stay and hastening trauma surgery in this group of patients may bring a wide range of benefits, both for the patient and the aiding institution.

Author Response

Dear Author, 

This manuscript aims to assess whether patients with proximal femur fracture on chronic antiplatelet therapy may benefit from early surgery by shortening of the admission-to-surgery interval. 

Introduction is thorough and it highlights current principles of perioperative antiplatelet therapy management in a certain group of patients.

Experimental section elaborates on every step of the study Including operative techniques and types of anesthesia.

Results comprise adequate graphs, but paging could be improved.

Answer: Thanks very much for your comments.

Discussion should include another study limitation, which is management of pediatric population potentially included in this field of interest. Here is an article about Outcome of pinning in patients with slipped capital femoral epiphysis: risk factors associated with avascular necrosis, chondrolysis, and femoral impingement, published in J Int Med Res, DOI 10.1177/0300060517731683.

Answer: The scope of the trial was not the pediatric population. This limitation is not applicable to our study. Furthermore, it is very unlikely that they receive chronically antiplatelet drugs.

This article brings valuable information in a certain field of interest. Shortening the hospital stay and hastening trauma surgery in this group of patients may bring a wide range of benefits, both for the patient and the aiding institution.

Answer: Thanks for your comments.

Round 2

Reviewer 2 Report

No suggestion for this form